# Review on Heat Generation of Rubber Composites

**DOI:** 10.3390/polym15010002

**Published:** 2022-12-20

**Authors:** Ying Liu, Wenduo Chen, Dazhi Jiang

**Affiliations:** 1School of Materials, Shenzhen Campus of Sun Yat-sen University, No. 66, Gongchang Road, Guangming District, Shenzhen 518107, China; 2School of Materials, Sun Yat-sen University, No. 135, Xingang Xi Road, Guangzhou 510275, China

**Keywords:** heat generation, heat build-up, rubber composites, temperature rise

## Abstract

Rubber composites are extensively used in industrial applications for their exceptional elasticity. The fatigue temperature rise occurs during operation, resulting in a serious decline in performance. Reducing heat generation of the composites during cyclic loading will help to avoid substantial overheating that most likely results in the degradation of materials. Herein, we discuss the two main reasons for heat generation, including viscoelasticity and friction. Influencing factors of heat generation are highlighted, including the Payne effect, Mullins effect, interface interaction, crosslink density, bond rubber content, and fillers. Besides, theoretical models to predict the temperature rise are also analyzed. This work provides a promising way to achieve advanced rubber composites with high performance in the future.

## 1. Introduction

Owing to the high elasticity, excellent damping properties, and low stiffness, rubber composites are extensively used in many fields, for example, vibration isolators, tires, sealing gaskets, and high-voltage outdoor insulators [1] However, in these fields, filled rubbers are subjected to a considerable magnitude and frequency cyclic loading, which would generate a large amount of heat inside. In every loading cycle, part of the energy is changed into heat. Owing to the low thermal conductivity of rubber compounds, the generated heat inside rubbers slowly transmits to the surrounding environment; hence, the temperature of the compound significantly increases [2]. During the loading process, owing to the self-heating of rubber and outside conditions, the temperature of rubber parts can reach up to 110 °C [3]. Many researchers have focused on thermal conductivity [4,5,6] aiming to reduce the temperature by increasing the thermal conductivity, but few works focus on heat generation, which is another important aspect of rubber life.

When force is applied, the rubber sample begins to distort, then the work is stored as heat inside the specimen, leading to an increase in temperature due to heat accumulation. The dynamical deformation can be divided into two processes: irreversible process and reversible process. The entropic elasticity of rubber can be used to represent reversible processes and during irreversible processes. Work is either transferred into the fluctuation of molecular motion within the rubber material, which cannot be recovered as mechanical force, or it runs to the external system as heat. Keizo Akutagawa and his coworkers used the first and second thermodynamic laws to explain this process of heat building up in rubber composites at dynamical deformation [7].

Heat accumulation not only greatly weakens the mechanical properties (e.g., stiffness and modulus) of the rubber materials, but also leads to molecular aging [8]. Heat generation in rubber can greatly limit the development of high-performance rubber products. The high temperature generated from heat accumulation would change a crosslinked structure in the rubber, such as reducing the number of sulfur atoms between crosslinked polymer chains, changing the percent of monosulfide, disulfide, and trisulfide bonds, and thus increasing the concentration of cyclic sulfide or the formation of conjugate alkenes [9]. The temperature rise can greatly influence the viscoelastic and mechanical properties of rubber composites. As such, high temperature can result in the shortening of fatigue life, material aging, and blowout of tires, which further cause resource wasting, environmental pollution, and safety problem. Moreover, high temperature generation means energy dissipation, power loss of tires, and high rolling resistance, which bring about high energy consumption and large carbon emission [10].

Currently, the samples are primarily separated into dumbbell-shaped samples in tension mode and cylindrical samples in compression mode for the fatigue temperature rise test. The standard testing procedure for the tensile mode is to equip a dynamic testing apparatus with a temperature environmental chamber and an infrared camera (Figure 1a), which will be used to record the temperature in real time and analyze the results. Cylindrical samples were tested in compression mode by dynamic compression temperature rise testing instrument (Figure 1b). The thermocouple was put at the bottom or in the center of the sample, which can immediately record the instantaneous temperature. 

## 2. Reasons for Heat Generation

The hysteresis of rubber composites is a result of the strain cannot follow simultaneously the change of stress. The hysteresis is caused by the thermal movement of the internal molecular chain segments under the external force. However, the polymer chains will engender friction during the process of motion. The polymer chains cannot get a new balance quickly owing to the action of internal friction, leading to the lagging of strain. Hysteresis produces internal resistance, for instance, mechanical loss caused by the friction generated by the movement between molecular chain segments, which consumes part of the energy stored in the materials. This kind of energy will be transformed into heat and released. Nevertheless, the polymer is a weak conductor; therefore, the heat cannot be completely emitted, leading to temperature rise and the shortening of material life [11]. The two mechanisms of the energy dissipation of the filled rubber are summarized in the following.

### 2.1. Viscoelasticity

Based on the polymer viscoelasticity theory [12], the transfiguration of rubber composites can be divided into two parts, the viscous part and elastic part, often denoted as a viscous dashpot and an elastic spring, respectively. When a rubber specimen is subjected to cyclic loading, some energy loss is finally transformed into heat, and the temperature rise occurs in the rubber composites.

For ideal springs, the stress and strain are in phase, but for a dashpot, the strain will lag by 90° phase angle radians with the stress. A viscoelastic material with the properties partly of a spring and partly of a dashpot is of an intermediate phase angle and certain energy losses of the elastomeric materials undergoing cyclic deformation. The energy dissipated in the material is ultimately converted into heat, which leads to an increase in operating temperature in rubber products [13]. The energy dissipation equation reveals that the linear link between dynamic lag loss (tan δ) and the loss of energy given in the form of temperature rise, or loss modulus (E″) [14],
(1)$$\Delta E\;=\;\frac{E\prime \prime }{\omega }\int_{0}^{\frac{2\pi }{\omega }}\omega^{2}\varepsilon_{0}^{2}\sin^{2}\omega \mathrm{tdt}={\pi \varepsilon }_{0}^{2}\;E\prime \prime \approx {\pi \varepsilon }_{0}\sigma_{0}\tan\delta$$
where ΔE is the energy dissipation, σ_0_ is the stress amplitude, t is time, ε_0_ is the strain amplitude, ω is the angular frequency, δ is the phase difference between strain and stress, E″ is the loss modulus, and tan δ is the dynamic lag loss. Evidently, ∆E is proportional to E″ for an equal strain. The basic equation of the heat generated per unit volume per cycle is expressed by Equation (2) [15,16],
(2)Q=ΔE⋅f
where Q is the heat generation rate, and f is the frequency of loading oscillation.

The amount of heat generated every cycle is decided by the amplitude $\varepsilon_{0}$ of the oscillation (Equation (2)). Hence, when rubber materials are suffered from oscillations of constant load amplitude, the stiffer rubber material will show less heat generation and temperature rise due to the lower amplitude of the oscillation. Besides that, for service conditions that impose a fixed amplitude of an oscillation, composites with little loss modulus can be much better.

The stress softening between the loading path and the unloading is hysteresis, which is also called hysteresis loss [17]. After unloading to a stress-free state, filled rubber specimens cannot recover to the original spot, which implies a remarkable permanent set [18,19].

Hysteresis loss for every cycle is proportional to the loss modulus under low strain [20],
∆E = (π/4) (DSA/100)^2^G′tan δ(3)
where G′ is the loss modulus and DSA is the double strain amplitude, also known as the peak-to-peak displacement in percentage. The loss modulus and the square of the strain amplitude determine the hysteresis loss [17].

Both the stored and dissipated energies are included in each cycle. When the molecular arrangement changes, the strain potential energy is stored and fully recovered upon unloading. So, the amount of energy dissipated in the period of oscillatory straining can be calculated for the whole period. Little changes in the loop area can have a great influence on the temperature increment in the composites [21],
(4)$$U_{d}={\pi E}_{2}\varepsilon_{0}^{2}$$
where U_d_ is the energy dissipated in the complete strain cycle per unit volume. E_2_ is the ratio of steady-state stress to strain when the phase lag between stress and strain is 90 degrees.

The highest temperature that the center of the rubber block will reach is [15],
(5)$$T_{m}=T+\frac{{\mathrm{fU}}_{d}H^{2}}{8K}$$
where K is the coefficient of thermal conduction of the rubber, H is the thickness of the rubber block, f is frequency, and T is kelvin temperature (absolute temperature). It is noticeable that the heat conduction acts a critical role in deciding temperature rise.

Consequently, the heat generation is not only related to the stress and deformation involved in the rubber material, but also to the frequency of the applied loading and the environmental temperature [8,22].

### 2.2. Friction

Owing to internal frictions between polymer chains, particles, and polymer–particle interaction, temperature rise in rubber samples is an aspect of viscoelasticity [10]. The friction between polymer chains varies according to interaction, crosslinking, network, and temperature. The relaxation of polymer chains, segments, and crosslinking points will occur in an uncrosslinked rubber system under dynamic stress. When the system is crosslinked, the movement of polymer chains is limited, and relaxation mainly occurs in chain segments and crosslinking points. The structural unit moves to a new location faster than the rate of vacancies generated by the thermal motion of surrounding molecules under external force. The orderly mechanical motion will change into random thermal motion, resulting in heat generation. Below the glass transition temperature (Tg), the motion of chain segments is frozen. Thus, heat generation mainly occurs when the temperature is higher than Tg; that is to say, the chain segments can move when the temperature is in the rubber platform, and the crosslinking points and entanglements can limit the relaxation of chain segments. The heat generation of an ideal crosslinked network is very low, but the actual rubber crosslinked network has some defects such as dangle chains, ring structures, and unrestricted winding nodes. At the same temperature, the actual crosslinked network with the same components has more defects and more heat generation. Increasing the crosslinking density can reduce the proportion of network defects, then the friction between chain segments and heat generation can be reduced.

Cheng et al. [23] prepared a new interface structure by one-step modification and co-precipitation method, which is composed of reduced graphene oxide (rGO)/N-tert-butyl-2-benzothiazole sulfonamide (NS)/natural rubber (NR) (NR/NS-rGO) with covalent bond connections. During dynamic loading, NS is a bridge to connect the GO and NR through chemical bonds, preventing the rubber macromolecules from sliding along the surface of the filler. Therefore, the friction among fillers is reduced, and so is the friction between the fillers and substrates. Correspondingly, heat generation is reduced.

Therefore, the heat generation in rubber is not only related to stress and deformation of the rubber, but also the frequency of the dynamic loads applied to the rubber and temperature at a certain time. Energy loss comes from the friction between polymer chains, fillers, and polymer–filler interaction. In one word, the root of heat generation is viscoelasticity, and friction is the essence [13,24].

## 3. Influencing Factors of Heat Generation

### 3.1. Payne Effect

The storage modulus of unfilled rubber depends on the temperature and frequency of dynamic loads, which have nothing to do with the deformation amplitude of the rubber. Instead, the storage modulus for filled rubber depends on dynamic deformation, and the storage modulus value reduces noticeably as strain amplitude increases. The presence of a filler network in rubber composites above the percolation threshold can be blamed for this behavior, known as the Payne effect [25].

The amplitude dependency of the dynamic properties of composites is known as the Payne effect, which relates to a nonlinear property in the storage modulus of materials that decreases with the amplitude of dynamic strain. ∆E′ = E′_0.1%_ − E′_10%_ represents the Payne effect [11].

The modulus of the filled nature rubber composites steeply drops with increasing strain and expresses a typical nonlinear behavior. The Payne effect is mainly bound up with the filler network produced in the rubber matrix. Taking silica filler as an example, the rubber trapped between the fillers would lose activity, and the filler–filler network is increased. Therefore, the effective volume of the filler would sharply increase upon filler networking. Nevertheless, when the natural astaxanthin modifies the filler, the distribution and dispersion of the silica in the rubber matrix are obviously improved because of the weakened polarity of the silica surface, and the Payne effect is lowered [26]. The Payne effect does not appear to affect the structural evolution of the filler agglomeration, as evidenced by the fact that the destruction and recovery of carbon agglomerates do not coincide with the emergence of the Payne effect [27].

### 3.2. Mullins Effect

The Mullins effect describes the stress reduction among the initial loading and successive reloading of the loaded rubber under quasi-static cyclic loading [18]. The Mullins effect [28] is caused by irreversible structure changes, such as the breaking of chain segments and the separation of fillers.

The addition of fillers evidently increases the viscoelastic character of the rubber matrix. The strain cannot catch up with the change in stress under the cyclic loading situation, which results in a clear hysteresis ring (Figure 2). The Mullins effect and filler network structure go hand in hand during the cyclic loading [29].

The loss modulus is decided by the loading cycle number, which can be ascribed to cyclic stress relaxation. This is owed to the chain slipping and migration of entanglement [30]. The decreasing of loss modulus is expressed as the dynamic softening effect by Li Fanzhu and his coworkers [10], which is similar to the Mullins effect, then the relationship is established between the loading cycle number and loss modulus to characterize the energy dissipation rate.

### 3.3. Interface Interaction

The interfacial properties between the filler and rubber matrix play an essential role in the properties of rubber composites [31]. The interfacial interaction between fillers and polymer chains is a key factor in enhancing rubber performance [32]. The polymer–filler interaction can be measured by the energy loss fraction of the polymers, tan δ. For the viscoelastic behavior, the relationship of the energy loss fraction, W, of the rubber composite is expressed as follows [33,34],
(6)$$W=\frac{\pi \tan\delta }{\pi \tan\delta +1}$$

The energy loss fraction W at the tan δ is given by the following equation,
(7)$$W=\frac{\left(1-C\right)W_{0}}{1-C_{0}}$$
where C is the volume fraction of the constrained region, (1 − C) is the fraction of the amorphous region, and W_0_ and C_0_ denote the energy fraction loss and volume fraction (C_0_ is taken to be 0) of the constrained region for rubber blends, respectively. It can be rearranged as below,
(8)$$C=1-\frac{\left(1-C_{0}\right)W}{W_{0}}$$

The polymer–filler interaction increases with the value of C.

One of the most important problems is the dispersibility of the filler particles in the rubber matrix. The surface energy of the filler has determined the wetting of the filler by the polymer matrix [25]. Phonon scattering at the filler–polymer interface can be minimized by the creation of a strong covalent bond between the modified fillers and the polymer [35]. The generated heat inside quickly transmits to the surrounding environment owing to the high thermal conductivity; hence, the temperature of the rubber composites significantly decreases.

Studies on the nonlinear viscoelasticity and thermal conductivity of the B_4_C/NR composites demonstrate that the well-maintained and highly thermally conductive channels are the primary cause of the increased thermal conductivity [36]. There are two main reasons for the addition of B_4_C significantly reducing the heat generation: (1) heat has been conducted outside more effectively due to the increased thermal conductivity, and (2) the weak interfacial activity and low surface area decrease the interfacial friction effectively [36].

To improve the bonding strength between polymer matrix and filler, hydroxyl telechelic natural rubber (HTNR) with hydroxyl-terminated groups has been introduced into silica-reinforced natural rubber by Katueangngan et al. [37]. The increase in temperature rising with silica loading is a result of the silica–silica network and the poor dispersion of filler particles in rubber composites. Nevertheless, the addition of HTNR significantly decreases temperature rise. Adding HTNR can improve the dispersion of silica and reduce the silica–silica network. Thus, the re-agglomeration of filler is decreased, resulting in less heat generation. Furthermore, the reduction in heat generation is also owing to strengthened rubber–filler interactions through the hydroxyl groups of HTNR interacting with the silanol groups on silica surfaces.

Nanospring-filled elastomer composites have been constructed for two types (Figure 3): nanosprings and polymer chains have been directly blended to obtain system I; system II, on the other hand, is a tri-block chain–nanospring–chain structure that self-assembles. The findings suggested that the permanent deformation of composites could be reduced by the chemical connections at the interface between nanosprings and polymer chains. The good dispersion and interfacial interaction between the nanosprings and the polymer chains can significantly decrease the hysteresis loss of composites [38].

The relationship of crosslink density and entanglement points with the thermogenesis properties of NR was proposed by Yue-Hua Zhan [39]. They found that the crosslink and entanglement points can hold the rubber chain to restrict the movement of the polymer chain and reduce heat generation.

Lai et al. used the latex compounding ingredients without a strong reducing agent to enhance the thermal conductivity of NR/graphene composite properties by the NR/rGO interaction [40].

### 3.4. Crosslink Density

When uncrosslinked rubber composites are stressed, molecular chains can easily slide past each other and disentangle. A few crosslinks can increase the molecular mass, have a broader molecular mass distribution, and create branched molecules. It is much harder for these branched polymers to disentangle, and hence, tensile strength of the crosslinked elastomer increases. When a rubber composite is deformed by an outside force, parts of the energy lied in the polymer chains and can be used as a driving force for fracture. The leaving energy is dissipated through polymer movement into heat and cannot break chains. High crosslinking may restrict chain motions, and the network cannot release much energy.

The heat build-up property of elastomers depends strongly on crosslink density. Hysteretic effects decrease monotonically with increasing crosslink density. There is an optimal crosslink density for practical use. High enough crosslink densities should prevent viscous flow failure, whereas low enough densities should prevent brittle failure [15]. 

Fang and coworkers [14] studied the crosslink density of rubber compounds filled with different amounts of Si-69. The networks among coupling agent, silica, and polymers become more and more intensive, which increases the crosslink density of rubber compounds. When the crosslink density is increased, the loss modulus and dynamic lag loss of the sample show a declining trend.

Heat build-up and crosslinking have an important relationship [41]. A higher degree of network stability usually causes lower heat generation. A low degree of heat build-up can be expected by the comparison of heat generation tests before and after aging.

### 3.5. Bound Rubber

The interaction between polymer and filler determined by the types and numbers of adsorption sites determines the interfacial structure between the two [42]. At room temperature, bound rubber is a term used to describe polymer chains that a suitable solvent cannot remove from suspensions. Bound rubber also serves as an indirect indicator of how the polymer and filler interact.

On the surface of fillers, the ‘‘bound rubber layer’’ is composed of neighboring constrained chains and adsorbed chains. The polymer mass has a great influence on the thickness of the layer [11]. Unperturbed chains and restricted chains are combined to create the “constrained rubber layer” on the external layer. The inclusion of filler particles does not fundamentally impair the mobility of the unaffected chains that surround the confined rubber layer, also known as bulk rubber. Divide the layers with clear limits to simplify (Figure 4). For bound rubber limiting the movement of polymer chains, the energy loss caused by the glass transition of polymer is reduced [24]. 

### 3.6. Fillers

Since most the rubber materials exhibit poor mechanical properties, the addition of high loading of fillers is mandatory to achieve satisfactory mechanical properties. However, the maximum amount of filler always brings about friction and temperature rise [43]. Filler size, loading, coupling agent, and synergistic effects all have an impact on how much heat is generated when fillers are added to rubbers.

#### 3.6.1. Effects of Filler Size and Loading

According to the report, the deformation and reorganization of the aggregated filler particles under cyclic deformation are responsible for the temperature rise; in other words, the better dispersion of fillers or little filler–filler network could bring about less heat generation.

Carbon black has been extensively applied as a reinforcing filler for polymer composites [44,45,46,47]. Wongwitthayakool et al. [48] used various carbon black addition and performance (such as specific surface area and structure) to prepare hydrogenated acrylonitrile butadiene rubber (HNBR) composites. The amplitude of the temperature rise is more pronounced in specimens with high specific surface area and carbon black structure, and the temperature rise is greatly increased with carbon black loading and surface area rising. Graphene has a large specific surface area and corrugated structure; therefore, it can obstruct the rubber macromolecules from slipping along the surface of graphene; thus, the intrinsic friction and energy loss are reduced, and the internal heat rise is decreased [49]. 

Nature rubber with low surface and high carbon black structures exhibit low temperature rise [50]. The influence of carbon black loading on the performance of NR composites is studied by Zhang et al. [51], especially the temperature rise. The content of bound rubber increases with an increase of carbon black loading, which enhances the inner rub between the filler and polymer. Additionally, the free volume decreases in the process of the test, then the temperature rise progressively increases with filler loadings. In general, when fillers are used to reinforce rubber, the higher the filler content, the higher the heat accumulation of rubber composites [23]. Uraiwan Sookyung prepared NR/organoclay nanocomposites with various organoclay loadings. The results make known that the temperature rise increased with organoclay loading [52]. In theory, monodispersed filler particles that do not participate in agglomeration can adsorb more polymer chains. Increasing the number of filler particles can cut down the distance between particles, which would help the formation of bridging chains [42]. Hysteresis loss increases with an increase in dangling chain ends ratio by comparing the hysteresis loss for different rubbers [8].

#### 3.6.2. Effects of Coupling Agent

The modified filler particle surfaces led to good distribution and compatibility among the matrix and the filler, which had a homogeneously dispersed system [35,53]. At the same filler loading, the better dispersion of the filler in the rubber matrix, the more filler–rubber interaction points will be formed [54].

Bo Yang et al. [55] developed silica/NR composites by co-modifying silica with dodecanol and the silane coupling agent KH-592 (Figure 5). The silane coupling agent KH-592 can create a bridge structure between the filler and matrix when a mercapto group is present, which enhances the interaction between the silica and NR matrix. As a result, there is less friction between the filler–filler and the filler–polymer interaction points, and the rubber composite material with low temperature rise is prepared.

Xiao et al. [56] first use a new silica modifier TWEEN-20 (Figure 6), which has four long arms made up of three polyether chains with a terminal hydroxyl group and a fatty chain. A hydrogen bond is formed by the oxygen on the polyether and the silanol groups on the silica surface, and a chemical reaction may occur between the terminal hydroxyl and the silanol groups without any volatile organic compounds (VOCs). Besides that, to get good compatibility with the rubber matrix, the long fatty chain undercut silica polarity, so that silica modified by TWEEN-20 can homogeneously disperse in the polymer matrix. Therefore, the addition of TWEEN-20 can significantly reduce the HBU of the composites.

A graphene oxide (GO)-supported vulcanization accelerator (NS-rGO) is prepared by Cheng et al. [23] through a reaction of the vulcanization accelerator GO and N-tert-butyl-2-benzothiazole sulfonamide (NS). The temperature rise of the sample only is 2.6 °C when the amount of NS-rGO is 0.42 vol%, and it decreased by 0.9 °C compared with the neat nature rubber. It is due to that the heat generation in rubber is mostly produced by the friction between networks, for example, polymer–polymer, filler–filler, and filler–polymer networks. Under dynamic loading, NS is the bridge to connect GO and nature rubber through the strong chemical bonds to prevent the NR from sliding along the surface of GO. The dynamic friction between the fillers is reduced, and the friction between fillers and matrixes is also reduced, as is the heat generation. It shows that the validly chemical bonds among the fillers and polymer chains can actually decrease the heat generation of rubber compounds.

#### 3.6.3. Synergistic Effect

Research has found that heat transfer can be significantly improved by introducing a combination of different kinds of filling materials, such as micron and nanometer fillers. Compared with a single size and shape, the synergistic effect of multiple structures makes it more concise to form effective thermal conductive pathways in a polymer matrix [57,58]. Hybridization of nanofillers is a new method to promote the uniform dispersion of nanofillers in polymer matrices. To get excellent properties of synergistic effects, at least two different fillers will be plugged into the polymer matrix [59,60]. Jafarpour et al. found that the hybridization of carbon nanotube (CNT) and nanodiamond (ND) enhanced the stability of individual nanoparticles in styrene-butadiene rubber (SBR) and the thermal conductivity of composites in a synergistic manner [60].

In SBR/CB composites, the disaggregation and re-agglomerate of CB cause friction among CB particles and bring about vast amounts of energy dissipation in the form of heat. The results have shown that the addition of MoS_2_ has a good function in improving the dispersity of CB. MoS_2_ is a good lubricant to reduce the friction coefficient between MoS_2_ sheets and CB. The addition of 3 phr MoS_2_ leads to a 10 °C decrease in the temperature rise of the composites [61].

Mohanty et al. [62] used clay partly to replace CB can increase the temperature rise of SBR/CB composites. With the replacement of CB by nanoclay, the possibility of CB–nanoclay interface formation will increase leading to the increase of the inter-filler friction, ultimately resulting in the loss of energy in form of heat.

Due to GO’s surfactant properties and the potent p–p interaction with CNTs and GO, Li et al. [63] mixed up the CNTs with GO aqueous solution to obtain stable hybrid suspensions. It turns out that the GO/CNTs hybrid fillers can produce a unique three-dimensional filler network in a natural rubber matrix. This unique filler network can be used as a sacrificial bond to utilize energy before the material breakdown, which significantly lowers temperature rise in GO/CNTs/NR composites.

Graphene can inhibit the “flow” of carbon black and reduces the strain phase lag and loss factor of the composites. Enhancing the dispersion of filler and interface interaction of the composite can bring about the interfacial stress transfer more quickly. Hence, the modified GO and CB-filled NR can show brilliant dynamic mechanical properties [11]. 

Wei et al. [29] used GO and CNT hybrid fillers to replace partial CB (Figure 7). Parts of CNT and CB are absorbed on the surface of the GO sheet so it can produce a hybrid network beneficially. The synergistic intercalation between GO, CNTs, and CB could arrest their re-agglomeration and form a developed network. Therefore, the hybrid fillers enhance the dispersity of CB in the NR matrix. The temperature rise of GO/CNTs/CB/NR composites is less than CB/NR, which indicates that the excellent dispersing performance of CB, and the effective filler network can effectively improve the thermal conductivity of NR composites, resulting in the less temperature rise compared to CB/NR. Therefore, with the hybrid fillers affiliating, the temperature rise of CB/NR composites is markedly decreased.

Some papers about temperature rise are summarized in Table 1. Most of the papers test the bottom of the sample to obtain the temperature rise as shown in Figure 8, and only a few researchers test the center point of the sample. The lowest temperature rise of 7 °C can be measured when the amount of filler is 3.5 phr, but the tensile strength is only 13.3 MPa [36]. 

To sum up, the kind of fillers, particle size, filler loading, synergistic effect, and coupling agent can have a great influence on HBU. To obtain composites with the lower temperature rise, lower Payne effect, lower Mullins effect, stronger interface interaction between fillers and polymer chains, much more bound rubber and higher crosslink density are required.

Heat built-up, the heat generated by internal friction in the rubber, is only related to rubber components and interactions between components. It is also related to the loading peculiarity, such as frequency, amplitude, time, etc.

Temperature rise is a comprehensive reflection of the heat transfer inside the specimen between the specimen and between the specimen with the surrounding air. Therefore, the temperature rise is related to the heat built-up and thermal conductivity of composites. Raising the thermal conductivity of rubber composites is profitable for the diffusion of heat and the decrease of the temperature rise [53]. As a result, whether the composites can dissipate heat in time is the major factor for the normal service of rubber products. The primary and internal factors that determine the thermal conductivity coefficient values of polymers and polymer composites are typically chain structures of polymer matrix materials, interfacial thermal resistances, and intrinsic thermal conductivity coefficient values of thermally conductive fillers [68]. The improvements in thermal conductivity of polymer materials are very important for their application as thermal interface materials (TIMs) [69,70] and electromagnetic screening materials [71]. 

## 4. Numerical Simulation of Temperature Rise

Generally, the method of the physical experiment is used by the industry to validate their rubber product design improvements. However, there are significant limitations of rubber testing with the experimental method. The experimental methods need additional equipment, floor area, and operators to arrange the testing setups. One method to solve these issues is to use properly validated numerical models of composites. Combining such models with design experiments will make it possible to gain the best operating and design parameters [72].

Wongwitthayakool obtained the correlation between viscoelastic results got from oscillatory Rubber Processing Analyzer (RPA) as a standard test and the temperature rise measured from a flexometer [48]. The temperature rise can thus be measured using the RPA data. The study built a relationship between filler characteristics, magnitude of reinforcement, viscoelastic, and temperature rise behaviors. As a standard test, the temperature rise (T_s_) is often measured from the high loading flexometer using the RPA results. tan δ is less useful as an indicator of HBU than E″.
T_s_ = 18.019 ln(E″) − 61.971(9)
T_s_ = 33.907 ln(tan δ) + 102.83(10)

The temperature rise is the result of energy dissipation, explained by the hysteresis that is the mechanical response under cyclic loadings. Its prediction requires the resolution of the mechanical and thermal equations. 

According to the report, there are three methods to solve this problem [19,73]:(I)Fully coupled algorithm. The mechanical and thermal equations are solved simultaneously.(II)Fractional step algorithm. The thermomechanical equation is divided into two easier equations that are solved separately (first solve the mechanical equation and then solve the thermal equation).(III)Uncoupled algorithm. It consists of deformation, dissipation, and thermal modules. First of all, solve one cycle of the mechanical equation to evaluate the dissipation. Then solve the thermal equation for many cycles on one fixed geometry until the rise of temperature is significant. After that, use the actual temperature to update the mechanical equation.

As a result of the low simulation cost, the uncoupled method is constantly being used to solve the thermomechanical coupling equation that occurs under cyclic loading. It is proved that the numerical results agreed with the experimental data very well. To simulate the appearance of the self-heating during the fatigue process, the finite element method (FEM) often be used [19].

### 4.1. Classical Model

Based on Equation (1), the model of the relationship between heat generation and crosslink density is created, along with heat generation and dynamic lag loss [14]. The dynamic characteristics are updated as a function of strain and temperature due to the modified Kraus model and iterative solution process. The influence of creep on deformation and dynamic softening on the loss model of rubber materials are taken into consideration [10]. The numerical results agree with the experimental data well. Therefore, it requires precise experimental dynamic material characteristics to achieve the exact prediction.

Saux’s group used a simple phenomenological approach to overcome the difficulties of the classical method based on thermodynamic models. The validity of the method is suitable both on the transient and stabilized temperature fields. This uncoupled approach is still suitable for a temperature rise of less than 20 °C [73].

In Li’s work [74], the transient temperature of rubber tires is performed according to the thermo–mechanical coupling approach and viscoelastic theory (Figure 9). They used nature rubber and carbon black N234 to make a solid rubber tire, then tested the tire by a rubber rolling test apparatus. The solid rubber tire was composed of two parts: metal rim and rubber tire. With the rotation cycle from start, the temperature rises rapidly. Owing to thermal balance, a platform emerges at the time of around 1000 s for the simulated results and about 2000 s for the experiment results. There is always a delay between the simulated data and experiment data. Latency increases as the rotational speed and compressive displacement increase.

Luo [17,75,76] deeply considered the influence of frequency and loading strain amplitude on temperature rise. The relationship among loss modulus, frequency, and strain of the composites is built up through the Kraus model ($E_{K}^{\prime \prime }$) and Maxwell model ($E_{M}^{\prime \prime }$).
(11)$$E_{K}^{\prime \prime }\left(\Delta \varepsilon \right)=E_{\infty }^{\prime \prime }+\frac{2\left(E_{m}^{\prime \prime }-E_{\infty }^{\prime \prime }\right){\left(\Delta \varepsilon ∕{\Delta \varepsilon }_{c}\right)}^{m}}{1+{\left(\Delta \varepsilon ∕{\Delta \varepsilon }_{c}\right)}^{2m}}$$
(12)$$E_{M}^{\prime \prime }=\sum_{i=1}^{N}\frac{E_{i}{\omega \tau }_{i}}{1+\omega^{2}\tau_{i}^{2}}$$

In Equation (11), ∆ε_c_ is the strain amplitude eigenvalue, E″_m_ is the maximum loss modulus. E″_∞_ is the loss modulus at higher strain amplitude, which reaches its asymptotic plateau. m is a non-negative phenomenological exponent with the value about 0.5. It is almost independent of frequency, temperature, and filler content. In Equation (12), τ_i_ and E_i_ are the relaxation time and the elastic modulus of the Maxwell element, respectively. The relationship between Δε and E″ can be expressed by the Kraus model, and the relationship between ω and E″ can be expressed by the Maxwell model, as shown in Equations (11) and (12). They are always used to describe the strain amplitude dependence and the frequency dependence behaviors of rubber composites, respectively.

The Kraus model is used to describe the Payne effect. It turns out that hysteresis loss showed an increasing tendency with the strain amplitude and frequency. The energy lost across the entire deformation cycle is frequently calculated using the viscoelastic model, which is related to hysteresis loss with strain amplitude and loss modulus [17]. NR/CB cylindrical specimens were passed the dynamic compression mode test (Figure 1b). A comparison between the simulation and test data of temperature rise was carried out. It turns out that (Figure 10), at the low-frequency condition, the simulation results and the experimental data do not match well [75].

The amplitude of the Payne effect is tended to rise with the temperature dropping. The hysteresis loss of the rubber material is tended to decrease with the increase of temperature. A method to forecast the hysteresis loss of the rubber material at various temperatures is devised and confirmed in light of the experimental data from the Payne effect and the Kraus model. The predicted hysteresis losses according to this model at different temperatures and strain amplitudes agreed with the experimental results well. In other words, if the relationship between the Payne effect and temperature is given, the hysteresis loss can be predicted in the condition of a known strain amplitude at an arbitrary temperature [76]. It is not enough to only consider the influence of temperature rise. To acquire more precise simulation data for the low-frequency loading and the large strain amplitude conditions, the dynamic property softening needs to be taken into consideration.

The heat generation and the hysteresis loss of rubber composites under different frequencies and surrounding temperatures are predicted based on the Maxwell model [8]. As the loading frequency rises, the loss factor of the rubber r composites rises as well, and the reliance of the frequency weakens as the temperature rises. A chief cause is that the motion of polymer chains cannot follow the change of external loading with high frequency, which brings about much more energy loss and internal friction in rubber composites. With the ambient temperature increases, the thermal movement of the chain segment tends to quicken, and the effect of frequency on the loss factor shows a decreasing tendency.

### 4.2. New Models

Khiêm et al. put forward a physically motivated constitutive model to describe the inelastic behavior of filled rubber composites in multiaxial states of deformation [18]. Therefore, the rubber networks are separated into two anisotropic damage networks (M and H) and an induced anisotropic elastic network (E). The M network linked polymer chains attached to the filler’s surface in irreversible adsorption, while the H network contains polymer chains attached to a reversible adsorption site (Figure 11). Fillers are regarded as rigid bodies in the damage networks. The Mullins effect explains the deformation caused by permanent damage in the M network, and the hysteresis is caused by the strain-induced network recovery in the H network. 

The study also compared the published available test data of filled silicone rubber composites. As seen in Figure 12, there is a little bit of a misconception about the model with the tested data.

Circular loading rubber composites display a complex response, The main characteristic is stress softening and hysteresis caused by fatigue and dissipative heating. According to the thermodynamic principles, Guo and his coworkers [21,77] presented a new thermo viscoelastic damage method to predict inelastic fatigue phenomena. Two kinds of dissipative network rearrangements were taken into consideration, containing the unrecoverable rearrangements and the recoverable rearrangements viscoelasticity inducing damage. This is supposing that the recoverable viscoelastic rearrangements are caused by the movement of non-entangled free chains and entangled chains superimposed on a neat elastic perfect rubber network (Figure 13).

They used styrene-butadiene rubber was filled with three different amounts of carbon black prepared dog-bone-shaped specimens. The dynamic stretching mode test (Figure 1a) results were compared with the constitutive model. The predictive model capabilities were checked from the comparison between the calculated data and the experiment data. This model can predict the temperature evolution, particularly for a dependence of temperature evolution on filler loading and pre-stretch level, which is in accordance with the result of intrinsic dissipation. However, this model is only useful for flat, thin specimens with a constant cross-section that is subjected to cyclic pre-stretched loads of constant amplitude.

According to the dynamic stretching mode test (Figure 1a), the comparison of the stress–strain hysteresis loop at the 250th cycle is shown in Figure 14a [77]. The temperature changes tested in the middle region of the sample surface are compared with the simulation results in Figure 14b. Although there is a slight discrepancy between the model demonstration and the tested results of the stress–strain curves, the temperature evolution is almost predicted by the model.

According to the second principle of thermodynamics, a thermo-visco-hyperelastic constitutive model is established to describe the self-thermal evolution of elastic materials under cyclic loading (Figure 15a). The constitutive model regards the stress–strain response as the result of two polymer networks acting in parallel. The model shows that the total of the overall resistance to deformation should balance with an equilibrium state A and a time-dependent deviation with regard to the equilibrium response B. A is the non-linear elastic spring and B is the non-linear elastic spring in series with a viscous dashpot [78].

This temperature rise will have a great effect on the constitutive stress–strain behavior by producing a thermal softening of the rubber composites. The thermo-mechanical model, which is used to explain temperature-related mechanical behavior, consists of a thermal resistance functioning in series with a mechanical resistance connecting to the stress-free thermal dilatation and the big strain rubber elastic behavior, respectively [79]. 

Taking account of the influence of filler on the thermo-mechanical response, introduce a tensile amplification factor into the model (Figure 15c) [80]. 

They created dog-bone-shaped specimens out of styrene-butadiene rubber that were filled with three different concentrations of carbon black. Then, the samples were tested by the dynamic stretching mode (Figure 1a). The predicted temperature evolutions due to the temperature rise of the filled rubbers are presented in Figure 16. The data showed that the model captures temperature evolution from temperature rise in a satisfactory manner. However, for large volumes of particulate fillers, it is obvious that there is a significant difference between the test results and simulation results. This difference is attributed to the influence of highly adding amounts on the model under large strains.

A three-element model was used to reproduce the stress–strain data with two rubber elastic elements and a viscous element [81]. The model considers three factors to predict the temperature changes at the adiabatic conditions: viscous dissipation effects, isentropic elastic, and entropic elastic, as shown in Figure 17 [81]. 

According to the model, the nominal stresses added to are equal to the total nominal stress. Besides that, the nominal stress is equal to the stretch ratio.

The upshot is thus that, in the case of small deformation, the is entropic elastic effect is dominant, while in the case of large deformation, the entropy elastic effect is dominant. Carbon black-filled styrene-butadiene rubber was selected for the test. The samples were tested by the dynamic stretching mode test. Figure 18 summarizes the relationship between the predicted and experimental temperature changes and the strain under different loading conditions. The calculated temperature changes are lower than the experimental ones. The author thought that there are two reasons. The first reason is the restriction of measuring and testing technique. It is rather difficult to record precise figures at very small deformation due to noise. There is a second reason for the discrepancy is the estimation shortage of the heat conduction. Actually, there is thermal convection and conduction with the surrounding air inside the sample and between the sample holder.

However, in order to use this technique on rubber composites, some questions need to be further investigated. For example, a quantitative interpretation of the temperature changes of different rubber composites under different loading modes is required. Heat conduction in the rubber composites and heat convection from the surface of the rubber composites to the outside air should be taken into consideration.

### 4.3. FEM

FEM is one of the simulation methods to solve differential equations. Temperature rise in rubber composites by FEM is conducive to guide the molecular structure and formula design of rubber composites.

A thorough understanding of the heat build-up and temperature rise in rubber through FEM is crucial because it can direct the design of rubber products’ structures as well as their materials [10]. Most of the current models are mainly based on the viscoelasticity of the rubber material [82,83].

Lin et al. [16] discussed the temperature distribution of the tires at different conditions, which is carried out by the heat generation calculation using the product of the hysteresis factor and the total strain energy.

A new prediction method of heat generation is proposed by Milan et al. [84]. The method includes the prediction of hysteresis loss. For example, the dissipated energy within the polymer composites is studied by a visco-plastic rubber constitutive model of FEM. The finite element model (Figure 19a) uses three-dimensional solid hexagonal mesh for preliminary mesh division and then refines the boundary surfaces of rubber parts. The rubber blend sold under the trade name TG-B-712 by Serbian producer “TIGAR technical rubber” was chosen. The caoutchouc-butyl rubber TG-B-712 contains 40 vol% of carbon black particles. The specimen was compressed using an eccentric mechanical press. Figure 19b shows the comparison of the temperatures in the center of the rubber specimen obtained by experiment and calculation. The discrepancies between the test data and the calculation results are obvious.

Ghosh et al. [85] employed the product of the elastic strain energy of the rubber composite and the tangent of the loss angle to assess the energy dissipation.

Futamura et al. [86] pointed out that the deformation index relates the type of deformations to the modulus effect on energy loss. The index can be determined based on experimental data of tire performance or FEM analysis. Using FEM, the deformation indices and the energy losses are calculated for all elements of a rolling truck tire. The deformation index concept is used to simplify the fully coupled iterative FEM approach into a noniterative computational method in thermomechanical analysis.

According to the thermal–mechanical coupling method and nonlinear viscoelastic theory, the temperature rise and temperature variation of rubber sample under constant dynamic displacement amplitude and static compression cyclic loading are predicted by finite element analysis as shown in Figure 20. NR/CB cylindrical specimens were prepared by a traditional two-roll open mill. According to the dynamic compression mode test (Figure 1b), they found that the effect of dynamic softening and creep on the viscoelastic properties are considered firstly [10]. On this basis, they also proposed a practical method to calculate the transient temperature distributions and rolling resistance by building a 2D axisymmetric model [74].

He and his coworkers compared the advantages and disadvantages of the Lagrangian model, the Eulerian model, and the Plane Strain model of tires at different boundary conditions. The Lagrangian finite element model has the best prediction accuracy and maximum computational time. The three models are roughly the same as the experimental data and simulation results [87].

## 5. Summary and Perspectives

This review systematically discusses three aspects: the reason for heat generation, the effects of heat generation, and the prediction of temperature rise. The heat generation is not only related to the stress and deformation involved in the rubber material, but also the frequency of the applied loading and the environmental temperature. Energy loss comes from the friction of polymer chains, nanoparticles, and polymer chain nanoparticles. In one word, the root of heat generation is viscoelasticity, but friction is the essence.

The filler has a great effect on heat generation. The kind of fillers, filler particle size, filler loading, synergistic effect, and coupling agent can have a powerful influence on heat generation. To obtain composites with lower temperature rise, lower Payne effect, lower Mullins effect, stronger interface interaction between fillers and polymer chains, much more bound rubber, and higher crosslink density are required.

Models to predict the temperature rise are also analyzed theoretically. Currently, the FEM is one of the most important numerical methods to predict temperature rise. Before the prediction, the properties of the prepared composites should be known.

The current models are mainly based on the viscoelasticity of the material. In particular, the temperature rise curve is carried out by calculating the energy loss in each cycle. The simulation result is often affected by filler content, strain, frequency, etc., which has a small deviation from the actual results. The root cause of heat generation is the friction coming from polymer chains, nanoparticles, and polymer chain nanoparticles. Therefore, it is recommended to start from the perspective of friction in future simulations, but still a long way.

## Figures and Tables

**Figure 1 polymers-15-00002-f001:**
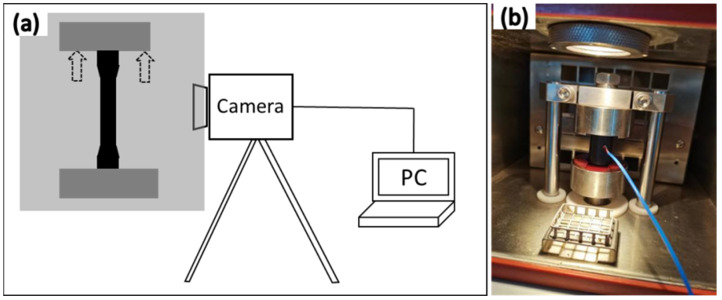
(**a**) Schematic diagram of infrared camera test. (**b**) Dynamic compression temperature rise experiment.

**Figure 2 polymers-15-00002-f002:**
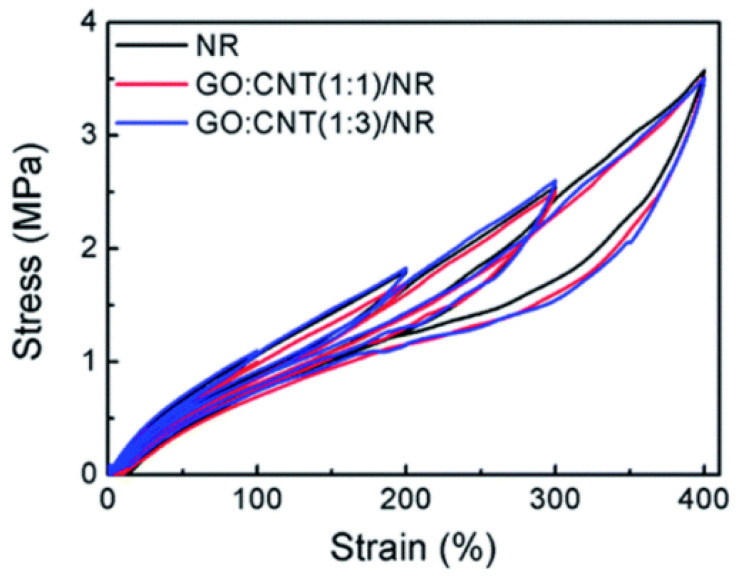
Hybrid filler–filled cyclic hysteresis curves of NR at various strains [29].

**Figure 3 polymers-15-00002-f003:**
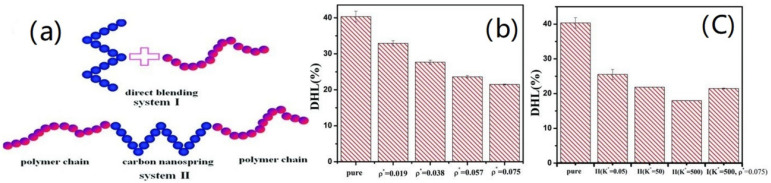
(**a**) Two systems filled with nanosprings. (**b**) Hysteresis loss of system I. (**c**) Hysteresis loss of system II [38]. The dimensionless spring constant is denoted by K* and ρ* represents the dimensionless interfacial chemical coupling density.

**Figure 4 polymers-15-00002-f004:**
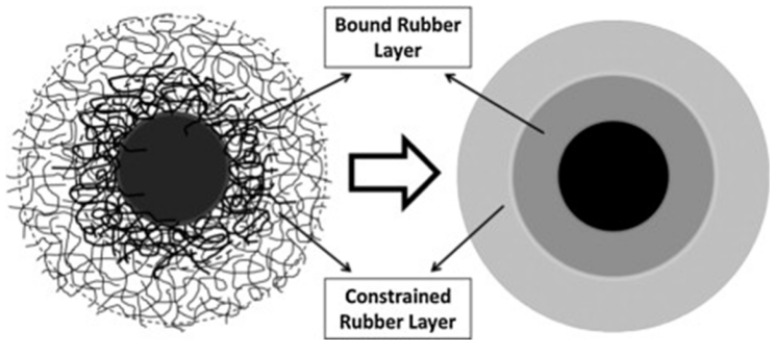
Schematic representation of interfacial structure between filler and polymer (bold lines mean adsorbed chains) [42].

**Figure 5 polymers-15-00002-f005:**
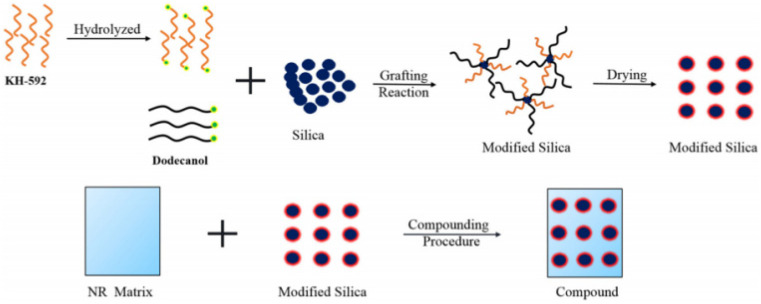
Flowchart for the production of silica/NR for co-modified silica particles with dodecanol and KH-592 [55].

**Figure 6 polymers-15-00002-f006:**
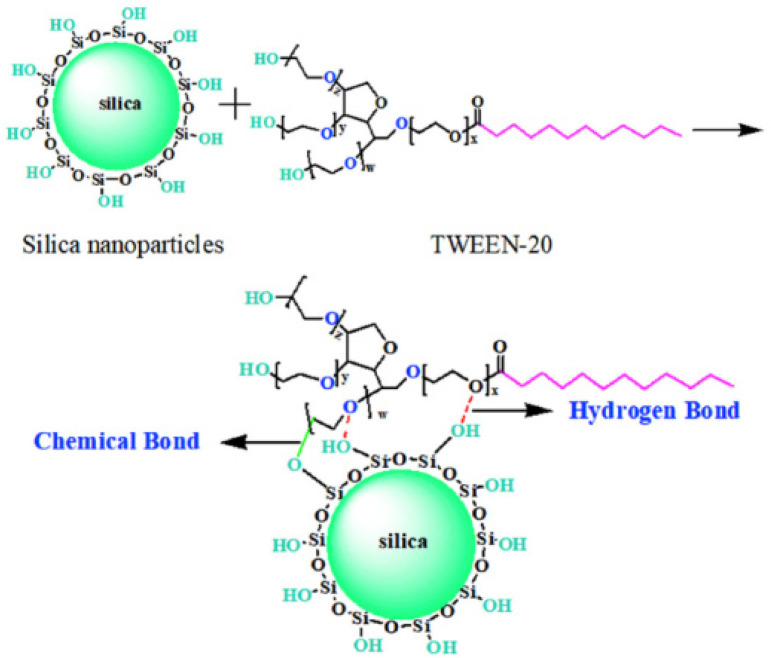
Schematic diagram of the interaction between TWEEN-20 with silica [56].

**Figure 7 polymers-15-00002-f007:**
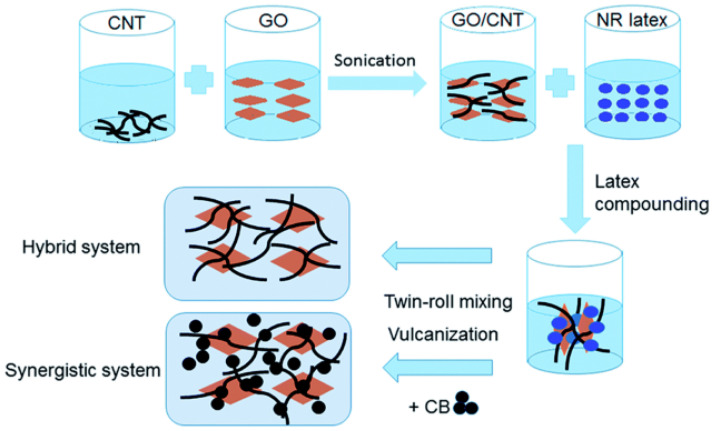
Latex mixing process utilized to create the GO/CNT/NR composites is depicted schematically [29].

**Figure 8 polymers-15-00002-f008:**
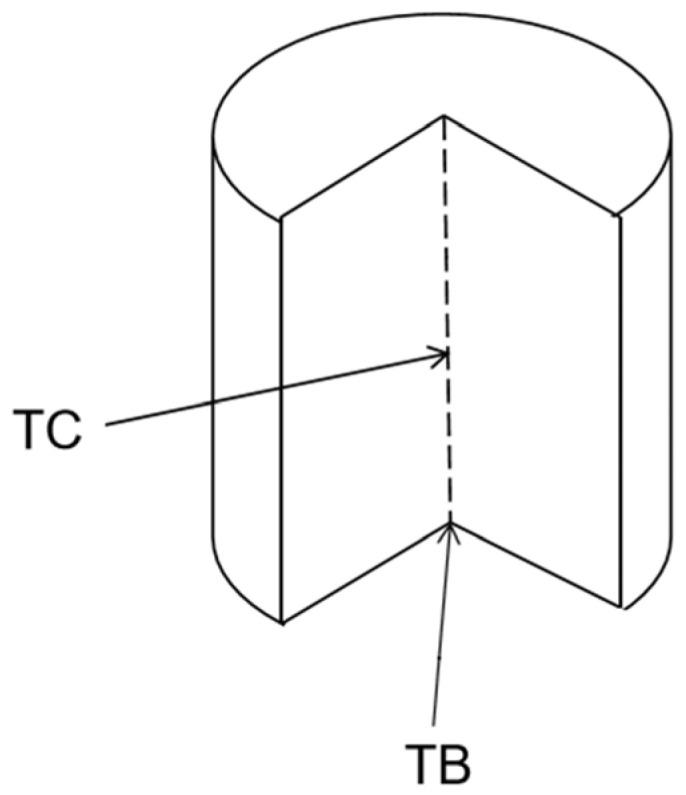
Description of test position.

**Figure 9 polymers-15-00002-f009:**
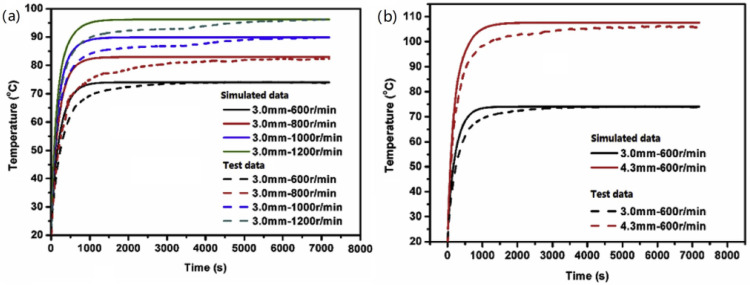
Comparison of temperature–time curves between experimental results and simulation results: (**a**) different rotating speed and (**b**) different compressive displacement [74].

**Figure 10 polymers-15-00002-f010:**
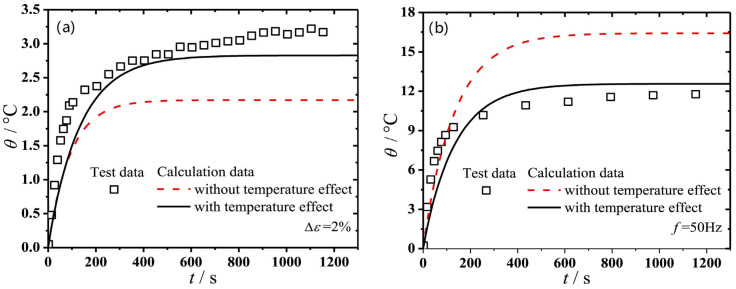
Comparison of the temperature rise results between simulation and test. (**a**) Frequency is 50 Hz and pre-strain 20%. (**b**) Strain amplitude is 5% and pre-strain 20% [75].

**Figure 11 polymers-15-00002-f011:**
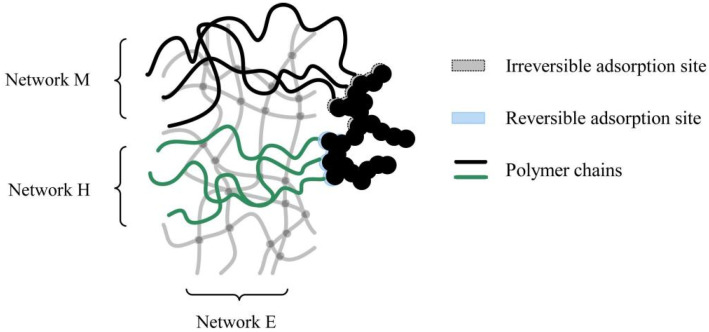
Two adsorption sites on the filler surface [18].

**Figure 12 polymers-15-00002-f012:**
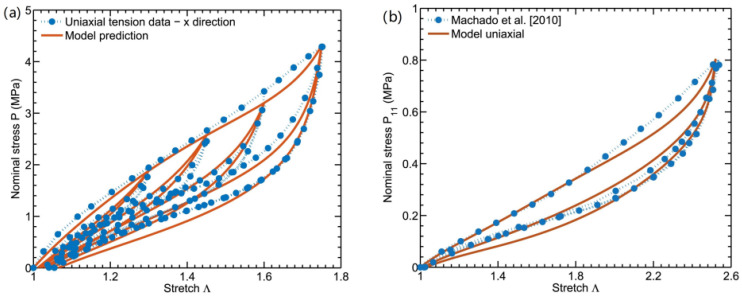
(**a**) Comparison of the experimental data and modeling findings for the cyclic uniaxial tension at various stretch ratios; (**b**) simulation and experimental results [18].

**Figure 13 polymers-15-00002-f013:**
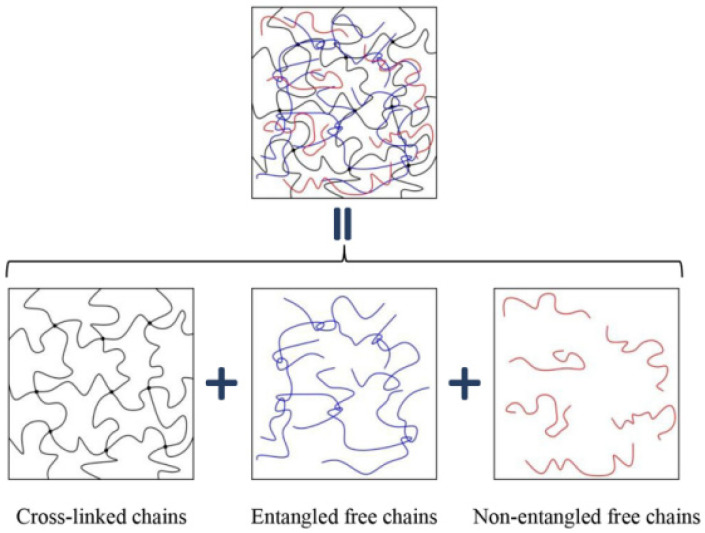
Rubber network is decomposed into three superposed chain populations [21].

**Figure 14 polymers-15-00002-f014:**
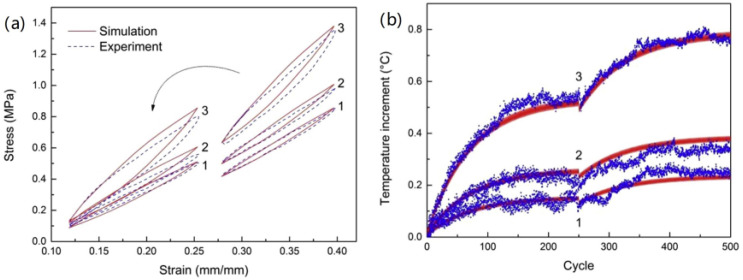
(**a**) Stress–strain curves of two blocks with different pre-tension levels at the 250th cycle. (**b**) Temperature evolution under two blocks with different pre-tension levels (simulation: red solid line, experiment: blue dot) [77].

**Figure 15 polymers-15-00002-f015:**
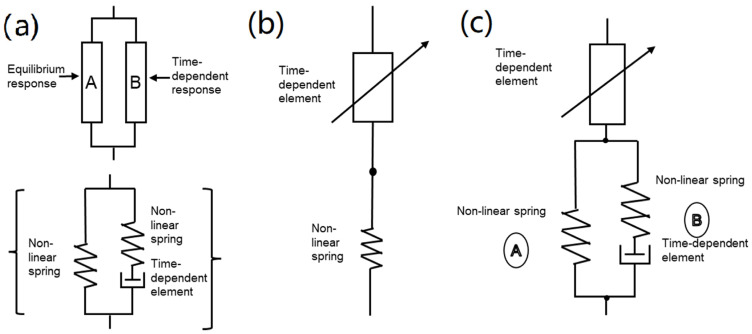
(**a**) Total deformation resistance is decomposed into the equilibrium resistance A and equilibrium response B [78], (**b**) schematic representation of thermo-visco-elastic model [79], and (**c**) schematic representation of thermo-visco-elastic model [80].

**Figure 16 polymers-15-00002-f016:**
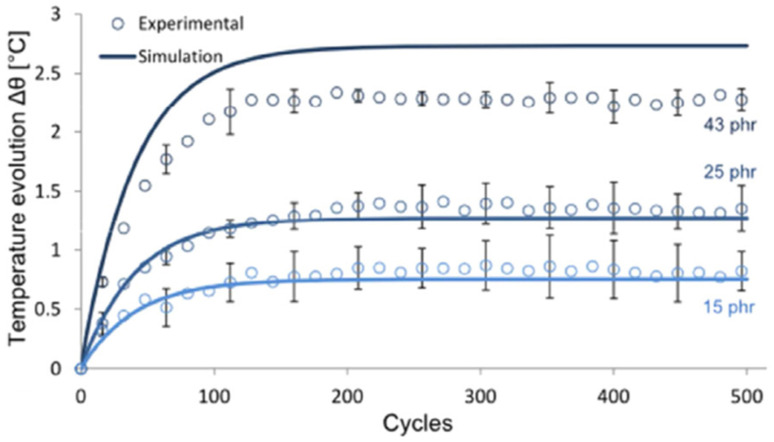
Predicted and experimental temperature rise [80].

**Figure 17 polymers-15-00002-f017:**
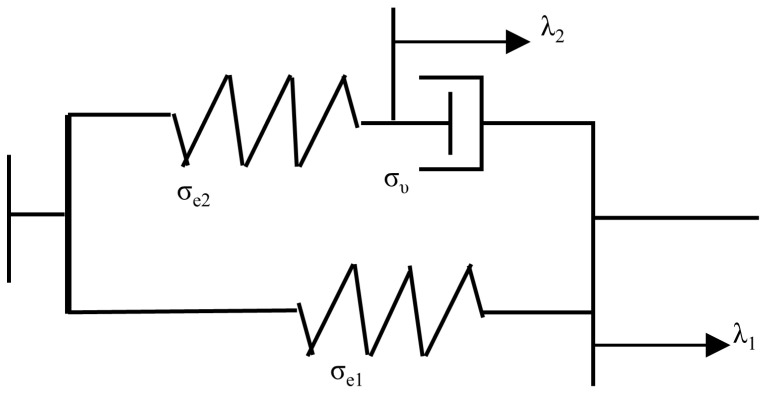
Material model [81].

**Figure 18 polymers-15-00002-f018:**
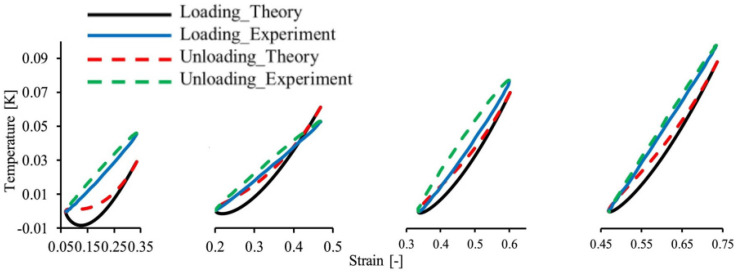
Temperature changes versus strain under different loading conditions [81].

**Figure 19 polymers-15-00002-f019:**
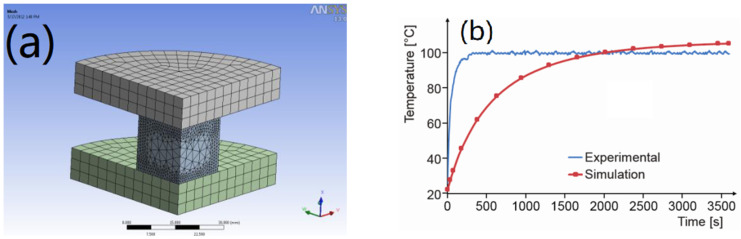
(**a**) FEM used in the verification process. (**b**) Comparison between experimental temperature and predicted temperature at the center of rubber specimen [84].

**Figure 20 polymers-15-00002-f020:**
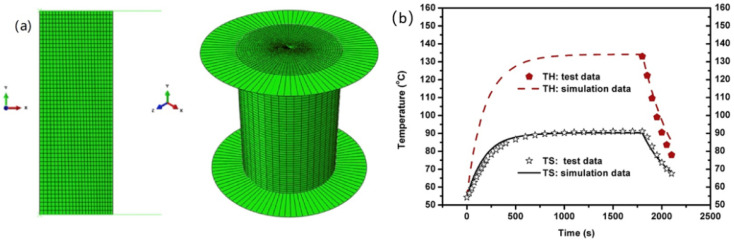
(**a**) Axisymmetric geometric model of the cylindrical rubber sample. (**b**) Temperature rise by considering the softening effect of the dynamic property [10].

**Table 1 polymers-15-00002-t001:** Temperature rise of rubber nanocomposites.

Polymer	Filler	Loading	Temperature Rise/°C	Tensile Strength/MPa	Test Position	Reference
NR	SiO_2_	50 phr	15.4	27.7	TC	[26]
NR	SiO_2_	70 phr	21.2	21	TC	[56]
NR	CB/SiO_2_	40/20 phr	22.7	18	TC	[64]
NR	SiO_2_	30 phr	13	21.5	TC	[55]
NR	B_4_C	14 vol%	7	13.3	TB	[36]
NR	CB	30 phr	11.5	18.5	TC	[51]
NR	CB/GO	40 phr	9	27	TC	[65]
NR	CB	60 phr	4.7	19	TC	[66]
NR	CB/MoS_2_	50 phr	36	23	TC	[61]
NR	rGO	0.42 vol%	4.6	-	TC	[23]
NR	CB/SiO_2_	25/25 phr	27	-	TC	[14]
NR	CB	42 phr	56.11	25.3	TC	[62]
NR	CB/GO/CNTs	-	15	-	TB	[29]
NR	GO/CNTs	0.5/3 phr	7	-	TB	[29]
NR	CB	40 phr	80	-	TB	[10]
NR	SiO_2_	30 phr	11.7	19.25	TC	[37]
NR	kaolin	6 phr	11	-	TC	[67]

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
