# Peer review of "Review on Heat Generation of Rubber Composites"

_polymers, 2022, doi:10.3390/polym15010002_

Round 1

Reviewer 1 Report

The manuscript reviews the mechanism and the parameters affecting the thermal behaviour of NR composites. This will be useful when these composites are used in the actual applications.

This is a review article. This is the first time that the topic has been addressed .

The manuscript deals with thermal behavior of rubber composites with various fillers. Each filler acts differently thus give varied thermal behaviors. The thermal changes directly affect the end use properties of the rubber composites.

The paper is well written. The readers can follow the article easily as it has been structured in a suitable format.

This is a review article. The readers will gain collective knowledge as the work is rationalized by several relevant research articles.

The references are appropriate and timely cited.

Author Response

We thank the referee for the very positive appraisal of our submission.

Reviewer 2 Report

Thank you very much for your reviewer articel. In article are more information on numerical simulation of propagation temperature.

could be explain better experimental methods that were using in simulation.

The articel requires minor formal adjustments, there are typos in it.

Author Response

To respond the reviewer’s comments clearly, we state our answers in two parts.

(1) Could be explain better experimental methods that were using in simulation.

Response: Thanks very much for this suggestion. The experimental methods as follow. 

 Currently, the samples are primarily separated into dumbbell-shaped samples in tension mode and cylindrical samples in compression mode for the fatigue temperature rise test. The standard testing procedure for the tensile mode is to equip a dynamic testing apparatus with a temperature environmental chamber and an infrared camera (Fig. 1a), which will be used to record the temperature in real time and analyze the results. Cylindrical samples were tested in compression mode by dynamic compression temperature rise testing instrument (Fig. 1b). The thermocouple was put at the bottom or in the center of the sample, which can immediately record the instantaneous temperature.

Figure 1. (a) Schematic diagram of infrared camera test. (b) Dynamic compression temperature rise experiment.

This paragraph had been added to the L70 of section “Introduction

According to the reviewer’s comments and suggestions, the following sentences had been added to the section “Numerical simulation of temperature rise” in the revised version:

L506: They used nature rubber and carbon black N234 to make a solid rubber tire, then tested the tire by a rubber rolling test apparatus. The solid rubber tire was composed of two parts: metal rim and rubber tire.

L534: NR/CB cylindrical specimens were passed the dynamic compression mode test (Fig. 1b). A comparison between the simulation and test data of temperature rise was carried out.

L590: They used styrene-butadiene rubber filled with three different amounts of carbon black prepared dog-bone shaped specimens. The dynamic stretching mode test (Fig.1a) results were compared with the constitutive model.

L631: They created dog-bone-shaped specimens out of styrene-butadiene rubber that were filled with three different concentrations of carbon black. Then the samples were tested by the dynamic stretching mode (Fig.1a).

L652: Carbon black filled styrene-butadiene rubber was selected for the test. The samples were tested by the dynamic stretching mode test.

L684: The rubber blend sold under the trade name TG-B-712 by Serbian producer "TIGAR technical rubber" was chosen. The caoutchouc-butyl rubber TG-B-712 contains 40 vol% of carbon black particles. The specimen was compressed using an eccentric mechanical press.

L703: NR/CB cylindrical specimens were prepared by a traditional two-roll open mill. According to the dynamic compression mode test (Fig. 1b), they found that the effect of dynamic softening and creep on the viscoelastic properties are considered firstly [10].

(2) The article requires minor formal adjustments, there are typos in it.

Response: Thanks very much for this suggestion. We have corrected some typos in the revised version. For example,

L88: a viscous part and an elastic part” has been changed to “viscous part and elastic part”.

L91: “occurred” has been changed to “occurs”.

L163: “At” has been changed to ”During”

…… 

Thank you for your comments and suggestions. Those comments are all valuable and very helpful for revising our paper, as well as the important guiding significance to our researches.
